# Beta-Blocker and Renin–Angiotensin System Inhibitor Combination Therapy in Patients with Acute Myocardial Infarction and Prediabetes or Diabetes Who Underwent Successful Implantation of Newer-Generation Drug-Eluting Stents: A Retrospective Observational Registry Study

**DOI:** 10.3390/jcm9113447

**Published:** 2020-10-27

**Authors:** Yong Hoon Kim, Ae-Young Her, Myung Ho Jeong, Byeong-Keuk Kim, Sung-Jin Hong, Seunghwan Kim, Chul-Min Ahn, Jung-Sun Kim, Young-Guk Ko, Donghoon Choi, Myeong-Ki Hong, Yangsoo Jang

**Affiliations:** 1Division of Cardiology, Department of Internal Medicine, Kangwon National University School of Medicine, Chuncheon 24341, Korea; hermartha1@gmail.com; 2Department of Cardiology, Cardiovascular Center, Chonnam National University Hospital, Gwangju 61469, Korea; myungho@chollian.net; 3Division of Cardiology, Severance Cardiovascular Hospital, Yonsei University College of Medicine, Seoul 03722, Korea; kimbk@yuhs.ac (B.-K.K.); HONGS@yuhs.ac (S.-J.H.); DRCELLO@yuhs.ac (C.-M.A.); kjs1218@yuhs.ac (J.-S.K.); ygko@yuhs.ac (Y.-G.K.); cdhlyj@yuhs.ac (D.C.); mkhong61@yuhs.ac (M.-K.H.); jangys1212@yuhs.ac (Y.J.); 4Division of Cardiology, Inje University College of Medicine, Haeundae Paik Hospital, Busan 48108, Korea; cloudksh@gmail.com

**Keywords:** beta-blocker, diabetes, myocardial infarction, outcomes, renin–angiotensin system inhibitor

## Abstract

Long-term clinical outcomes in patients with acute myocardial infarction (AMI) and prediabetes or diabetes who received ß-blockers (BB) and renin–angiotensin system inhibitor (RASI) therapy after successful newer-generation drug-eluting stent (DES) implantation are limited. We compared the two-year clinical outcomes in such patients. A total of 9466 patients with AMI in the Korea AMI Registry were classified into six groups according to their glycemic status and presence or absence of BB + RASI therapy: normoglycemia and BB + RASI users (*n* = 2217) or nonusers (*n* = 243), prediabetes and BB + RASI users (*n* = 2601) or nonusers (*n* = 306), and diabetes and BB + RASI users (*n* = 3682) or nonusers (*n* = 417). The primary endpoint was major adverse cardiac events (MACEs) defined as all-cause death, recurrent myocardial infarction (Re-MI), or any repeat revascularization, and the secondary endpoint was the cumulative incidence of hospitalization for heart failure (HHF). In patients with BB + RASI, despite similar primary and secondary clinical points between the prediabetes and diabetes groups, the cumulative incidence of Re-MI (adjusted hazard ratio: 1.660; 95% confidence interval: 1.000–2.755; *p* = 0.020) was higher in the diabetes group than in the prediabetes group. In all three different glycemic groups, BB + RASI users showed reduced MACEs, cardiac death, and HHF compared to those of BB + RASI nonusers. In this retrospective observational registry study, BB + RASI therapy showed comparable clinical outcomes except for Re-MI between prediabetes and diabetes in patients with AMI during a two-year follow-up period.

## 1. Introduction

Even though there is some debate on the efficacy of ß-blockers (BB) in patients with acute myocardial infarction (AMI) [1,2,3], oral BB is being prescribed in patients with AMI and reduced left ventricular ejection fraction (LVEF) to decrease mortality and major adverse cardiac events (MACE) according to the recommendations of current guidelines in the absence of contraindications [4,5,6,7]. For the same reason, renin–angiotensin system inhibitors (RASI) are being prescribed in patients with AMI and reduced LVEF [4,5,6,7]. Additionally, some recent reports have suggested that BB and RASI are effective in patients with preserved LVEF [8,9] Hence, among the many patients with AMI who do not have a reduced LVEF and/or heart failure (HF), oral BB is administered in real-world practice. However, BB has been known to increase the risk of severe or prolonged hypoglycemia and delay recovery from hypoglycemia in patients with diabetes [10]. Currently, the presence or absence or the degree of adverse cardiovascular risk associated with BB use in patients with diabetes is debatable [11,12]. Prediabetes is also associated with cardiovascular morbidity and mortality [13,14], and the number of individuals with prediabetes could exceed 470 million by 2030 [15]. However, despite their important effects on cardiovascular disease, research concerning the long-term clinical outcomes of prediabetes in patients with AMI is more limited than those in diabetes. In Korea, more than 40% of AMI patients received BB + RASI therapy after drug-eluting stent (DES) implantation [16]. Therefore, we investigated the two-year major clinical outcomes between prediabetes and diabetes mellitus (DM) in patients with AMI who received BB + RASI therapy after successful percutaneous coronary intervention (PCI) with newer-generation DES to estimate the clinical effects of prediabetes in such patients.

## 2. Materials

### 2.1. Study Design and Population

In the current study, we attempted to confine to type 2 DM (T2DM) patients for diabetes. We considered T2DM based on a previous study [17], which included patients from the Korea AMI Registry (KAMIR). Hence, a total of 21,343 AMI patients aged ≥30 years at the onset of diabetes who underwent successful PCI using newer-generation DESs from November 2005 to June 2015 in the KAMIR [18] were evaluated. The types of newer-generation DESs are given in Table 1. KAMIR [18] was designed to capture real-world treatment practices and the short- and long-term outcomes of AMI patients, evaluate the current epidemiology and analyze the prognostic factors of AMI, and improve the long-term prognosis of individual patients. Eligible patients were ≥18 years of age at the time of hospital admission. Patients with incomplete laboratory results, including unidentified results of blood hemoglobin A1c and blood glucose (*n* = 8314, 39.0%), patients who were lost to follow-up (*n* = 1067, 5.0%), and patients who received BB (*n* = 1423, 6.7%) or RASI alone (*n* = 1073, 5.0%), were excluded. After exclusion, a total of 9466 patients with AMI who underwent successful PCI using newer-generation DES were included. The patients were classified into normoglycemia (*n* = 2460, 26.0%), prediabetes (*n* = 2907, 30.7%), and diabetes (*n* = 4099, 43.3%) groups. These patients were divided into six groups according to their glycemic status and the presence or absence of BB + RASI therapy on admission: normoglycemia (BB + RASI users (*n* = 2217, group A1) and BB + RASI nonusers (*n* = 243, group A2)), prediabetes (BB + RASI users (*n* = 2601, group B1) and BB + RASI nonusers (*n* = 306, group B2)), and diabetes (BB + RASI users (*n* = 3682, group C1) and BB + RASI nonusers (*n* = 417, group C2)) (Figure 1, Table 1, Appendix A). All 9466 patients finished a two-year clinical follow-up through face-to-face interviews, phone calls, or medical chart reviews. All clinical events were evaluated by an independent event adjudicating committee. The processes of event adjudication have been described in a previous publication of the KAMIR investigators [18]. This was a nonrandomized, multicenter, observational retrospective cohort study. The ethics committee at each participating center approved the study protocol, and all participants provided informed consent before inclusion in the study. The study was conducted according to the ethical guidelines of the 1975 Declaration of Helsinki.

### 2.2. Percutaneous Coronary Intervention Procedure and Medical Treatment

Diagnostic coronary angiography and PCI were done through either the femoral or radial artery using standard techniques [19]. Aspirin (200–300 mg) and clopidogrel (300–600 mg) were administered before PCI, Alternatively, ticagrelor (180 mg) or prasugrel (60 mg) was administered. After PCI, the total duration of dual antiplatelet therapy (DAPT, a combination of aspirin (100 mg/day) with clopidogrel (75 mg/day), ticagrelor (90 mg twice daily), or prasugrel (5–10 mg/day)) was recommended for >12 months. Moreover, in this registry, there were no operator restrictions for performing PCI.

### 2.3. Study Definitions and Clinical Outcomes

AMI was defined according to current guidelines [4,5]. A successful PCI was defined as a residual stenosis <30% and thrombolysis in myocardial infarction grade 3 flow for the infarct-related artery (IRA) after the procedure. Glycemic categories were determined based on the glycosylated hemoglobin (HbA1c), fasting plasma glucose (FPG), and random plasma glucose (RPG) levels of the patients at the index hospitalization as well as their medical history. Diabetes was defined as either known diabetes for which patients received medical treatment (insulin or antidiabetics) or newly diagnosed diabetes defined as an HbA1c level ≥6.5%, FPG ≥126 mg/dL (7.0 mmol/L), and/or RPG ≥200 mg/dL (11.1 mmol/L) according to the American Diabetes Association clinical practice recommendations [20]. Prediabetes was defined as an HbA1c of 5.7–6.4% and an FPG of 100–125 mg/dL (5.6–6.9 mmol/L) [20]. Furthermore, the estimated glomerular filtration rate (eGFR) was calculated using the Modification of Diet in Renal Disease (MDRD) study equation [21]. The primary clinical endpoint of this study was the occurrence of MACEs defined as all-cause death, recurrent myocardial infarction (Re-MI), or any repeat revascularization. The secondary endpoint was the cumulative incidence of hospitalization for HF (HHF) during a two-year follow-up period. All-cause death was classified as cardiac death (CD) or non-CD. Any repeat revascularization was composed of target lesion revascularization, target vessel revascularization (TVR), and non-TVR.

### 2.4. Statistical Analysis

For continuous variables, differences among the three groups were evaluated using an analysis of variance or the Jonckheere–Terpstra test, while post hoc analysis was performed using the Hochberg test or Dunnett T3 test; the data are expressed as the mean ± standard deviation. For categorical variables, intergroup differences were analyzed using a χ^2^ test or Fisher’s exact test, as appropriate. Data are expressed as counts and percentages. Various clinical outcomes were estimated using the Kaplan–Meier method, and intergroup differences were compared using a log-rank test. To determine meaningful variables, all variables with *p* < 0.1 were included in the univariate analysis. After univariate analysis, variables with *p* < 0.001 and known conventional risk factors of poor outcomes in the AMI population were considered potential confounding factors and were entered into the multivariate analysis. These variables included the following: male, age, LVEF (≤40%), body mass index, diastolic blood pressure, ST-elevation myocardial infarction (STEMI), hypertension, dyslipidemia, previous MI, previous PCI, previous cerebrovascular accidents, current smokers, creatine kinase myocardial band (CK-MB), N-terminal pro-brain natriuretic peptide (NT-ProBNP), serum creatinine, eGFR (<60 mL/min/1.73 m^2^), total cholesterol, triglyceride, high-density lipoprotein (HDL) cholesterol, low-density lipoprotein (LDL) cholesterol, clopidogrel, ticagrelor, cilostazole, calcium channel blockers, lipid-lowering agents, left anterior descending coronary artery (LAD; IRA), right coronary artery (RCA; IRA and treated vessel), one-vessel disease, ≥3-vessel disease, stent diameter, and number of stents (Table 2). In the same way, the comparisons of major clinical outcomes between BB + RASI users and nonusers in all three groups (normoglycemia, prediabetes, and diabetes) were performed as shown in Table 3. For all analyses, two-sided *p* < 0.05 were considered statistically significant. All statistical analyses were performed using SPSS version 20 (IBM; Armonk, NY, USA).

## 3. Results

### 3.1. Baseline Characteristics

The mean LVEF value of this study population was more than 50% (Table 1). Group A1 (normoglycemia and BB + RASI users) had the most men, the highest number of one-vessel disease; the highest systolic and diastolic blood pressure, peak CK-MB, and eGFR; the highest prescription rate of ticagrelor, angiotensin-converting enzyme inhibitors (ACEIs), prasugrel, and lipid-lowering agents; the highest LAD as IRA; and the largest diameter of deployed stents. Group B1 (prediabetes and BB + RASI users) had the highest number of current smokers; the highest levels of total and low-density lipoprotein cholesterol; and the highest use of intravascular ultrasound. Group C1 (diabetes and BB + RASI users) had the highest number of non-STEMI (NSTEMI), multivessel disease, and RCA as IRA and treated vessel; the highest total number of deployed stents; the oldest mean patient age of all groups; the lowest level of eGFR and high-density lipoprotein cholesterol; and the highest number of risk factors for coronary heart disease (e.g., hypertension, dyslipidemia, previous history of MI, PCI, coronary artery bypass graft, stroke, and HF), as expected. Moreover, the highest levels of NT-ProBNP, high sensitivity C-reactive protein, and triglycerides were also observed in group C1. Baseline characteristics of the total study population and BB + RASI nonusers are shown in Appendix A.

### 3.2. Clinical Outcomes

Table 2 and Table 3 and Figure 2 show the cumulative incidences of major clinical outcomes during the two-year follow-up period. As can be seen in Table 2, the cumulative incidences of MACEs (adjusted hazard ratio (aHR): 1.127; 95% confidence interval (CI): 0.860–1.477; *p* = 0.387), all-cause death, CD, any repeat revascularization, and HHF (aHR: 1.428; 95% CI: 0.760-2.685; *p* = 0.268) were similar between group B1 (prediabetes and BB + RASI users) and C1 (diabetes and BB + RASI users). However, the cumulative incidence of Re-MI was higher in group C1 than group B1 (aHR: 1.660; 95% CI: 1.000–2.755; *p* = 0.020). The cumulative incidence of MACEs (aHR: 1.464; 95% CI: 1.022–2.096; *p* = 0.038) was higher in group B1 than in group A1 (normoglycemia and BB + RASI users). The cumulative incidences of MACEs (aHR: 1.587; 95% CI: 1.139–2.012; *p* = 0.006) and Re-MI (aHR: 2.275; 95% CI: 1.218–4.247; *p* = 0.010) were significantly higher in group C1 than in group A1. In BB + RASI nonusers, the cumulative incidences of MACEs (aHR: 1.708; 95% CI: 1.126-2.590; *p* = 0.012), all-cause death (aHR: 1.759; 95% CI: 1.109–2.788; *p* = 0.016), and CD (aHR: 1.844; 95% CI: 1.106–3.075; *p* = 0.019) were higher in group C2 (diabetes) than in group A2 (normoglycemia). In Table 3, in all three groups (normoglycemia, prediabetes, and diabetes), BB + RASI therapy reduced the cumulative incidences of MACEs, all-cause death, CD, and HHF. Additionally, in the prediabetes group, BB + RASI users showed lower cumulative incidence of any repeat revascularization than that of BB + RASI nonusers. In the diabetes group, BB + RASI users showed lower cumulative incidences of Re-MI and any repeat revascularization than those of BB + RASI nonusers. Independent predictors for MACEs and HHF in BB + RASI users at two-year follow-up are shown in Appendix A. Male sex, STEMI, decreased LVEF (<40%), lipid-lowering agents, decreased eGFR (<60 mL/min/1.73 m^2^), ≥3-vessel disease, the use of intravascular ultrasound (IVUS), and ≥30 mm length of the deployed stent were meaningful independent predictors for MACEs. Moreover, old age (≥65 years), STEMI, decreased LVEF, decreased eGFR, and ACC/AHA type B2/C lesions were independent predictors for HHF in this study.

## 4. Discussion

Because fewer data are available in patients with prediabetes concerning their long-term prognosis, the authors wanted to examine whether or not long-term outcome in patients after AMI and prediabetes or diabetes who received BB and RASI and were treated with newer-generation drug-eluting stents would show differences in clinical outcomes after two years.

In this retrospective observational registry study, our analysis showed the following findings after BB + RASI therapy: (1) The cumulative incidences of primary and secondary clinical outcomes were similar between the prediabetes and diabetes groups. However, the cumulative incidence of Re-MI was higher in the diabetes group than in the prediabetes group. (2) The cumulative incidence of MACEs of the prediabetes group and the cumulative incidences of MACEs and Re-MI of the diabetes group were higher than in the normoglycemia group. (3) In all three different glycemic groups (normoglycemia, prediabetes, and diabetes), the cumulative incidences of MACEs, all-cause death, CD, and HHF were lower compared to those of who did not receive BB + RASI therapy. (4) The cumulative incidence of any repeat revascularization of the prediabetes group and Re-MI and any repeat revascularization of the diabetes group were also decreased after BB + RASI therapy than those of patients who did not receive BB + RASI therapy. (5) Old age, STEMI, decreased LVEF, and decreased eGFR were common independent predictors of both MACEs and HHF.

According to the European Society of Cardiology (ESC) guidelines [5], the use of BB in patients with STEMI has recently decreased from class I to class IIa. Moreover, some studies have raised questions regarding the mortality risk reduction capability of BB in patients with coronary artery disease [22,23]. However, BB has been considered as a core drug for secondary prevention in patients with coronary and other atherosclerotic vascular disease, including patients with AMI and T2DM, in the absence of contraindications [24,25]. RASI has been shown to have diverse beneficial effects on cardiovascular outcomes through the enhancement of endothelial function, cardiovascular remodeling, and progression of atherosclerosis [26,27]. Recent [12] and previous reports [28] have suggested that BB + RASI therapy is more effective for patients after AMI than BB or RASI alone to reduce mortality. Therefore, in our study, we compared the clinical outcomes in patients with AMI who received or did not receive BB + RASI therapy and observed that BB + RASI therapy reduced the cumulative incidences of MACEs, all-cause death, CD, and HHF compared to BB + RASI nonusers regardless of the glycemic status. These beneficial effects of BB + RASI therapy were more prominent during the first three months after discharge (Figure 2). In their study of 2679 patients with AMI and preserved LVEF, Puymirat et al. [29] demonstrated that 30-day mortality was significantly lower in patients who received BB during the first 48 h after admission compared to those who did not (HR: 0.46; 95% CI: 0.26–0.82; *p* = 0.008). Regarding more recently published results [30], RASI has also been shown to have early mortality reduction benefit in patients with diabetes and prediabetes.

In the Hoorn study [31], the risk of a recurrent cardiovascular event was similar between the normoglycemia and prediabetes groups. However, individuals with diabetes had an increased risk of recurrent cardiovascular events compared to individuals with normoglycemia during a median 4.1 years of follow-up after the first event. According to a recent report [30], after adjustment, the cumulative incidence of Re-MI of the diabetes group in RASI users was significantly higher than that of the prediabetes group (aHR: 1.999; 95% CI: 1.153–3.467; *p* = 0.014). Deedwania et al. [32] suggested that the association between diabetes and Re-MI may be a direct effect of diabetes. Hyperglycemia has been linked to an increase in plasma renin activity with RAS activation, which is known to impair insulin signaling [33]. Hyperglycemia, insulin resistance, and advanced glycation end-products can contribute to vascular inflammation and endothelial dysfunction in patients with diabetes [34]. Increased platelet activation, the presence of a chronic hypercoagulable state, and impaired fibrinolysis are also considerable factors in this unfavorable clinical outcome [35].

In the current study, despite the number of patients with decreased LVEF being the highest in the diabetes group (Table 1, Appendix A), the cumulative incidence of HHF was not significantly different among the three different glycemic groups regardless of the presence or absence of BB + RASI therapy after adjustment (Table 2). A possible explanation for this result is supported by the current guidelines that emphasize the beneficial role of BB or RASI in patients with AMI and reduced LVEF [4,5,6,7]. Moreover, these results may be similar to those of the CAPRICORN (Carvedilol Post-Infarct Survival Control in LV Dysfunction) randomized trial [36], which showed that BB + RASI therapy reduced mortality in patients with AMI and reduced LVEF (≤40%) with or without HF. Additionally, María et al. [37] demonstrated that, among hospitalized survivors of AMI, the use of RASI was associated with a lower risk of follow-up HF in patients with a LVEF ≤40% but not in those with a LVEF >40%. As shown in Appendix A, decreased LVEF (<40%) was a significant independent predictor for both MACEs (aHR: 1.634; 95% CI: 1.302–2.018; *p* < 0.001) and HHF (aHR: 6.923; 95% CI: 4.897–9.832; *p* < 0.001) in our study.

Even though there are diverse kinds of newly developed antidiabetic drugs and many efforts to prevent and delay the progression of disease in patients with prediabetes, intensive lifestyle changes and metformin therapy are the only universally accepted interventions for diabetes prevention [38]. BB and RASI are well-known drugs that can reduce short- and long-term morbidity and mortality in patients with AMI regardless of their glycemic status. However, current guidelines [4,5,6,7] do not suggest indications or have not shown the effects of these drugs in patients with AMI and prediabetes, and other reports regarding these comparative results between prediabetes and diabetes in such conditions have not been reported. Therefore, we believe that the results of this study can provide important insights for interventional cardiologists concerning the effects of BB + RASI therapy in patients with AMI and prediabetes and the clinical implication of prediabetes in patients with AMI. In this study, 11,877 patients (55.6%) were excluded due to incomplete laboratory results, including unidentified results of blood HbA1c and blood glucose, in addition to those excluded due to loss at follow-up and receiving BB or RASI alone. This appears to be a rather large proportion and can introduce selection bias if those who did not have HbA1c and blood glucose tests during hospitalization had a different clinical profile than those who had these tests. As mentioned, the baseline (e.g., sex difference, proportion of STEMI, degree of LVEF, and renal function) and lesion and procedural characteristic (e.g., extent of coronary artery disease and length of deployed stent) of the excluded patients could affect the main results of our study. Moreover, the ratio of three groups (normoglycemia, prediabetes, and diabetes) of the excluded patients may act as an important determinant of major clinical outcomes. Therefore, considering the composition of this study population, caution is needed in interpreting the results. Even though the size of the study population may be insufficient for conclusion, in this nationwide, retrospective, observational, multicenter registry analysis, more than 50 community and teaching hospitals in South Korea participated. However, larger, well-designed randomized controlled trials are required to confirm these results focusing on newer-generation DESs.

This study has some limitations. First, this study was based on a registry that was voluntary at each participating center, and the follow-up data were partially incomplete. Hence, the outcomes and conclusions are subject to the constraints inherent in these types of analyses. Second, the definition of patients with prediabetes based on their HbA1c levels rather than on an oral glucose tolerance test may have been a source of bias. Third, this study was based on discharge medications, and the KAMIR did not include detailed, whole data concerning the prescription doses, long-term adherence, discontinuation, and drug-related adverse events. The degree of glycemic control of the participants during the follow-up period was not clearly defined, which may constitute an additional bias. A majority of patients following AMI, especially with reduced LVEF, receive specific BB and ACEI or angiotensin II receptor blocker (ARB) and mineralocorticoid inhibitor. There are numerous clinical trials [36,39,40,41] with specific drugs in different configurations in patients after MI. However, because of the above limitations, we could not provide detailed information regarding specific drugs in different configurations in our enrolled patients. This is another major limitation of this study. Fourth, another weakness of this study was the lack of information about the BB and RASI drugs used. Different medicines from these two classes have different impacts on glucose metabolism. Therefore, it is important to clearly state which drugs/classes of drugs were taken into analysis and to check their impact on long-term outcomes considering their mechanistic influence on carbohydrate metabolism. However, because this registry did not include this information, we could not provide meaningful results concerning the relationship between these two class of drugs and their impact on glucose metabolism. Additionally, the heterogeneity of applied therapy may influence long-term clinical outcome of this study. Moreover, it is important to have data on the reason why BB or RASI were not used in patients in the BB + RASI nonuser group. Unfortunately, this information was not available in our registry data [42,43]. This point is also a big weakness of this study. Fifth, although we performed a multivariate analysis to strengthen our results, variables not included in KAMIR may have affected the study outcomes. Sixth, due to the lack of information concerning the occurrence of hypoglycemia during BB therapy in KAMIR, we could not provide data related to potential hypoglycemia in BB-treated patients, and the present analysis will not answer the question of whether or not BB and RASI may be indicated in patients with normal LVEF post-AMI. Finally, the two-year follow-up period in this study was relatively short to determine the long-term clinical outcomes of these strategies.

## 5. Conclusions

In conclusion, in this study, prediabetic and diabetic patients were not different in their outcome when treated with BB and RASI over two years, but they were certainly both better off than the corresponding groups without treatment with these established drugs post-AMI. However, the results should be interpreted with caution due to the lack of details on the adherence and feedback to the therapy prescribed. Moreover, further well-designed studies are warranted to confirm these results.

## Figures and Tables

**Figure 1 jcm-09-03447-f001:**
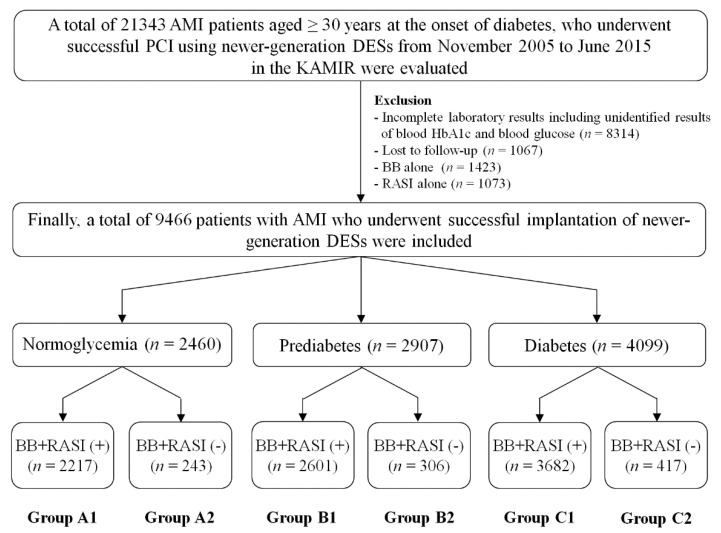
Flow chart showing the patient selection process for the study. AMI, acute myocardial infarction; PCI, percutaneous coronary intervention; DESs, drug-eluting stents; KAMIR, Korea AMI Registry; BB, ß-blocker; RASI, renin–angiotensin system inhibitor.

**Figure 2 jcm-09-03447-f002:**
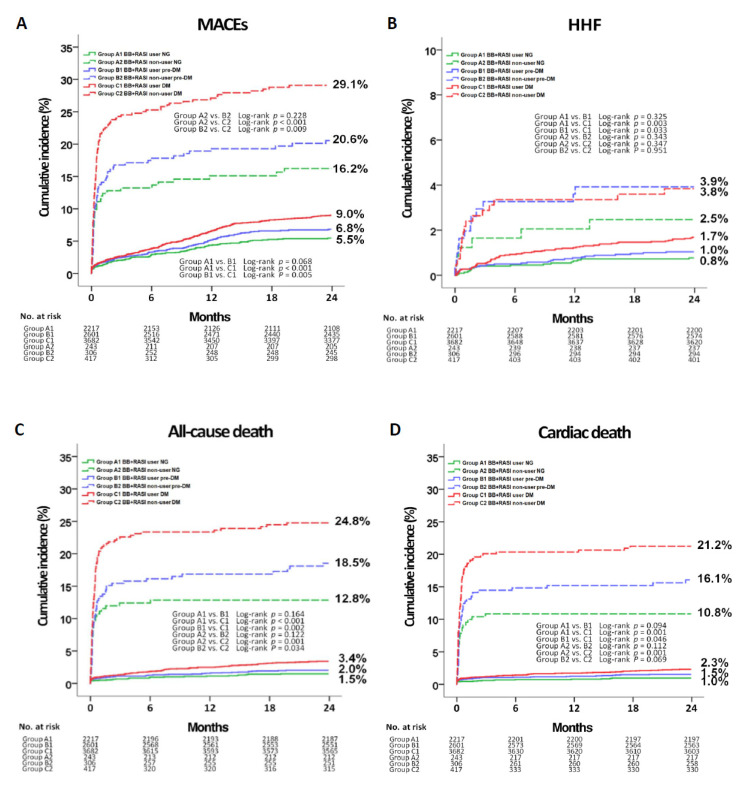
Kaplan-Meier analysis for the MACEs (**A**), hospitalization for HF (**B**), all-cause death (**C**), cardiac death (**D**), Re-ME (**E**), any repeat revascularization (**F**) in BB + RASI users and non-users. BB, ß-blockers; RASI, renin-angiotensin system inhibitor; NG, normoglycemia; Pre-DM, prediabetes, DM, diabetes; MACES, major adverse cardiac events; HHF, hospitalization for heart failure; Re-MI, recurrent myocardial infarction.

**Table 1 jcm-09-03447-t001:** Baseline characteristics of BB + RASI users.

Variables	Normoglycemia (Group A1, *n* = 2217)	Prediabetes (Group B1, *n* = 2601)	Diabetes (Group C1, *n* = 3682)	*p* Value			
				Group A1 vs. Group B1	Group A1 vs. Group C1	Group B1 vs. Group C1	Group A1 vs. Group B1 vs. Group C1
Male, *n* (%)	1803 (81.3)	1995 (76.7)	2588 (70.3)	<0.001	<0.001	<0.001	<0.001
Age (years)	60.4 ± 13.0	62.6 ± 12.4	63.7 ± 11.5	<0.001	<0.001	<0.001	<0.001
LVEF (%)	53.2 ± 10.3	53.2 ± 10.4	51.7 ± 11.3	0.968	<0.001	<0.001	<0.001
≤40%, *n* (%)	182 (8.2)	231 (8.9)	492 (13.4)	0.409	<0.001	<0.001	<0.001
BMI (kg/m^2^)	23.9 ± 3.1	24.4 ± 3.2	24.5 ± 3.2	<0.001	<0.001	0.212	<0.001
SBP (mmHg)	134.1 ± 27.7	131.6 ± 27.6	132.8 ± 27.9	0.002	0.095	0.090	0.009
DBP (mmHg)	82.3 ± 16.6	80.2 ± 16.5	79.7 ± 16.2	<0.001	<0.001	0.212	<0.001
STEMI, *n* (%)	1346 (60.7)	1578 (60.7)	1952 (53.0)	0.975	<0.001	<0.001	<0.001
Primary PCI, *n* (%)	1303 (96.8)	1530 (97.0)	1884 (96.5)	0.813	0.651	0.464	0.754
NSTEMI, *n* (%)	871 (39.3)	1023(39.3)	1730 (47.0)	0.975	<0.001	<0.001	<0.001
PCI within 24 h	774 (88.9)	887(86.7)	1480 (85.5)	0.154	0.019	0.398	0.064
Hypertension, *n* (%)	930(41.9)	1163 (44.7)	2210 (60.0)	0.054	<0.001	<0.001	<0.001
Dyslipidemia, *n* (%)	192(8.7)	294 (11.3)	529 (14.4)	0.002	<0.001	<0.001	<0.001
Previous MI, *n* (%)	6 (2.9)	55 (2.1)	166 (4.5)	0.085	0.002	<0.001	<0.001
Previous PCI, *n* (%)	84 (3.8)	117 (4.5)	277 (7.5)	0.220	<0.001	<0.001	<0.001
Previous CABG, *n* (%)	5 (0.2)	3 (0.1)	22 (0.6)	0.483	0.046	0.002	0.003
Previous CVA, *n* (%)	97 (4.4)	123 (4.7)	275 (7.5)	0.580	<0.001	<0.001	<0.001
Previous HF, *n* (%)	9 (0.4)	17 (0.7)	44 (1.2)	0.324	0.002	0.036	0.003
Current smokers, *n* (%)	1028 (46.4)	122 (47.2)	1464 (39.8)	0.559	<0.001	<0.001	<0.001
Peak CK-MB (mg/dL)	134.4 ± 208.2	142.0 ± 205.6	101.2 ± 137.9	0.204	<0.001	<0.001	<0.001
Peak troponin-I (ng/mL)	46.6 ± 81.5	49.8 ± 138.4	45.6 ± 144.8	0.381	0.751	0.296	0.515
Blood glucose (mg/dL)	136.6 ± 49.9	146.5 ± 45.7	222.8 ± 96.3	<0.001	<0.001	<0.001	<0.001
Hemoglobin A1c (%)	5.33 ± 0.43	5.96 ± 0.21	7.81 ± 2.87	<0.001	<0.001	<0.001	<0.001
NT-ProBNP (pg/mL)	264.0 (62.0–1225.0)	223.0 (56.0–919.0)	348.5 (78.8–1750.5)	0.215	<0.001	<0.001	<0.001
hs-CRP (mg/dL)	5.7 ± 31.2	8.9 ± 55.9	9.1 ± 43.6	0.030	0.003	0.888	0.047
Serum creatinine (mg/L)	0.99 ± 0.87	1.02 ± 1.02	1.13 ± 1.08	0.379	<0.001	<0.001	<0.001
eGFR (mL/min/1.73 m^2^)	92.1 ± 33.9	90.4 ± 37.3	86.6 ± 44.6	0.107	<0.001	<0.001	<0.001
<60 mL/min/1.73 m^2^	242 (10.9)	349 (13.4)	779 (21.2)	0.009	<0.001	<0.001	<0.001
Total cholesterol (mg/dL)	183.0 ± 40.1	190.6 ± 42.9	180.6 ± 48.9	<0.001	0.050	<0.001	<0.001
Triglyceride (mg/L)	120.8 ± 91.8	138.3 ± 111.8	154.5 ± 138.6	<0.001	<0.001	<0.001	<0.001
HDL cholesterol (mg/L)	44.5 ± 144	43.2 ± 15.8	42.0 ± 14.2	0.153	<0.001	<0.001	<0.001
LDL cholesterol (mg/L)	116.4 ± 36.1	122.6 ± 49.3	111.4 ± 38.8	<0.001	<0.001	<0.001	<0.001
Diabetes management							
Diet			314 (8.5)				
Oral agent			2305 (62.6)				
Insulin			173 (4.7)				
Untreated			890 (24.2)				
Discharge medications							
Aspirin, *n* (%)	2205 (99.5)	2587 (99.5)	3669 (99.6)	0.989	0.281	0.269	0.451
Clopidogrel, *n* (%)	1774 (80.0)	2229 (85.7)	3166 (86.0)	<0.001	<0.001	0.747	<0.001
Ticagrelor, *n* (%)	281 (12.7)	235 (9.0)	282 (7.7)	<0.001	<0.001	0.051	<0.001
Prasugrel, *n* (%)	147 (6.6)	122 (4.7)	194 (5.3)	0.003	0.030	0.319	0.010
Cilostazole, *n* (%)	304 (13.7)	495 (19.0)	749 (20.3)	<0.001	<0.001	0.210	<0.001
CCBs, *n* (%)	86 (3.9)	96 (3.7)	219 (5.9)	0.733	0.001	<0.001	<0.001
Lipid-lowering agents	2267 (93.7)	2431 (93.5)	3320 (90.2)	0.755	<0.001	<0.001	<0.001
IRA							
Left main, *n* (%)	37 (1.7)	31 (1.2)	65 (1.8)	0.178	0.837	0.076	0.117
LAD, *n* (%)	1152 (52.0)	1337 (51.4)	1707 (46.4)	0.699	<0.001	<0.001	<0.001
LCx, *n* (%)	371 (16.7)	439 (16.9)	638 (17.3)	0.894	0.568	0.642	0.815
RCA, *n* (%)	657 (29.6)	794 (30.5)	1272 (34.5)	0.501	<0.001	0.001	<0.001
Treated vessel							
Left main, *n* (%)	55 (2.5)	62 (2.4)	108 (2.9)	0.827	0.326	0.207	0.349
LAD, *n* (%)	1342 (60.5)	1577 (60.6)	2155 (58.5)	0.945	0.129	0.095	0.159
LCx, *n* (%)	563 (25.4)	666 (25.6)	1039 (28.2)	0.867	0.018	0.022	0.020
RCA, *n* (%)	783 (35.3)	961 (36.9)	1560 (42.4)	0.241	<0.001	<0.001	<0.001
ACC/AHA lesion type							
Type B1, *n* (%)	294 (13.3)	352 (13.5)	458 (12.4)	0.782	0.359	0.202	0.404
Type B2, *n* (%)	767 (34.6)	853 (32.8)	1284 (34.9)	0.187	0.829	0.087	0.205
Type C, *n* (%)	983 (44.3)	1145 (44.0)	1635 (44.4)	0.825	0.961	0.763	0.953
Extent of CAD							
1-vessel, *n* (%)	1219 (55.0)	1396 (53.7)	1605 (43.6)	0.384	<0.001	<0.001	<0.001
2-vessel, *n* (%)	655 (29.5)	785 (30.2)	1215 (33.0)	0.631	0.006	0.018	0.008
≥3-vessel, *n* (%)	343 (15.5)	420 (16.1)	862 (23.4)	0.552	<0.001	<0.001	<0.001
IVUS	452 (20.4)	617 (23.7)	766 (20.8)	0.006	0.702	0.007	0.006
OCT	16 (0.7)	27 (1.0)	24 (0.7)	0.245	0.746	0.093	0.215
FFR	25 (1.1)	36 (1.4)	54 (1.5)	0.428	0.294	0.830	0.544
Drug-eluting stents							
ZES, *n* (%)	696 (31.4)	861 (33.1)	1217 (33.1)	0.861	0.187	0.967	0.349
EES, *n* (%)	1108 (50.0)	1322 (50.8)	1871 (50.8)	0.557	0.533	0.993	0.792
BES, *n* (%)	368 (16.6)	360 (13.8)	500 (13.6)	0.008	0.002	0.767	0.003
Others, *n* (%)	45 (2.0)	58 (2.2)	94 (2.6)	0.690	0.215	0.453	0.406
Stent diameter (mm)	3.16 ± 0.42	3.16 ± 0.42	3.10 ± 0.42	0.686	<0.001	<0.001	<0.001
Stent length (mm)	27.5 ± 11.7	26.8 ± 11.2	27.5 ± 11.9	0.028	0.863	0.023	0.040
Number of stents	1.42 ± 0.74	1.46 ± 0.78	1.55 ± 0.82	0.029	<0.001	<0.001	<0.001

Values are mean ± SD or numbers and percentages or median (quartiles 1–3). The *p* values for continuous data obtained from the analysis of variance. The *p* values for categorical data from chi-square or Fisher’s exact test. LVEF, left ventricular ejection fraction; BMI, body mass index; SBP, systolic blood pressure; DBP, diastolic blood pressure; STEMI, ST-elevation myocardial infarction; NSTEMI, non-STEMI; PCI, percutaneous coronary intervention; MI, myocardial infarction; CABG, coronary artery bypass graft; CVA, cerebrovascular accident; HF, heart failure; CK-MB, creatine kinase myocardial band; NT-ProBNP, N-terminal probrain natriuretic peptide; hs-CRP, high-sensitivity C-reactive protein; eGFR, estimated glomerular filtration rate; HDL, high-density lipoprotein; LDL, low-density lipoprotein; CCBs, calcium channel blockers; IRA, infarct-related artery; LAD, left anterior descending coronary artery; LCx, left circumflex coronary artery; RCA, right coronary artery; CAD, coronary artery disease; ACC/AHA, American College of Cardiology/American Heart Association; IVUS, intravascular ultrasound; OCT, optical coherence tomography; FFR, fractional flow reserve; ZES, zotarolimus-eluting stent; EES, everolimus-eluting stent; BES, biolimus-eluting stent.

**Table 2 jcm-09-03447-t002:** Comparison of clinical outcomes according to the presence or absence of BB + RASI therapy at two-year follow-up.

	BB + RASI (+)						
	Group A1	Group B1		Unadjusted		Adjusted ^a^	
	Normoglycemia (*n* = 2217)	Prediabetes (*n* = 2601)	Log-Rank	HR (95% CI)	*p* value	HR (95% CI)	*p* value
MACEs	109 (5.5)	166 (6.8)	0.068	1.252 (0.983–1.594)	0.069	1.464 (1.022–2.096)	0.038
All-cause death	30 (1.5)	50 (2.0)	0.164	1.377 (0.876–2.166)	0.166	1.331 (0.647–2.040)	0.438
Cardiac death	20 (1.0)	38 (1.5)	0.094	1.582 (0.921–2.719)	0.097	1.987 (0.791–4.988)	0.144
Re-MI	30 (1.5)	40 (1.6)	0.691	1.101 (0.686–1.767)	0.691	1.288 (0.638–1.998)	0.479
Any repeat revascularization	58 (3.0)	83 (3.5)	0.376	1.164 (0.832–1.627)	0.376	1.583 (0.979–2.558)	0.061
Hospitalization for HF	17 (0.8)	27 (1.0)	0.325	1.355 (0.738–2.485)	0.327	1.129 (0.501–2.348)	0.769
	**Group A1**	**Group C1**		**Unadjusted**		**Adjusted ^a^**	
	**Normoglycemia (*n* = 2217)**	**Diabetes (*n* = 3682)**	**Log-Rank**	**HR (95% CI)**	***p* value**	**HR (95% CI)**	***p* value**
MACEs	109 (5.5)	305 (9.0)	<0.001	1.636 (1.315–2.036)	<0.001	1.587 (1.139–2.012)	0.006
All-cause death	30 (1.5)	117 (3.4)	<0.001	2.282 (1.528–3.409)	<0.001	1.579 (0.857–2.010)	0.143
Cardiac death	20 (1.0)	79 (2.3)	0.001	2.321 (1.421–3.792)	0.001	1.550 (0.698–1.992)	0.282
Re-MI	30 (1.5)	86 (2.6)	0.016	1.661 (1.096–2.517)	0.017	2.275 (1.218–4.247)	0.010
Any repeat revascularization	58 (3.0)	148 (4.5)	0.010	1.488 (1.098–2.015)	0.010	1.532 (0.973–2.412)	0.065
Hospitalization for HF	17 (0.8)	62 (1.7)	0.003	2.203 (1.288–3.768)	0.004	1.625 (0.806–3.275)	0.175
	**Group B1**	**Group C1**		**Unadjusted**		**Adjusted ^a^**	
	**Prediabetes (*n* = 2601)**	**Diabetes (*n* = 3682)**	**Log-Rank**	**HR (95% CI)**	***p* value**	**HR (95% CI)**	***p* value**
MACEs	166 (6.8)	305 (9.0)	0.005	1.313 (1.087–1.587)	0.005	1.127 (0.860–1.477)	0.387
All-cause death	50 (2.0)	117 (3.4)	0.002	1.667 (1.197–2.321)	0.002	1.391 (0.808–2.307)	0.234
Cardiac death	38 (1.5)	79 (2.3)	0.046	1.479 (1.004–2.177)	0.048	1.142 (0.608–2.144)	0.679
Re-MI	40 (1.6)	86 (2.6)	0.024	1.536 (1.056–2.235)	0.025	1.660 (1.000–2.755)	0.020
Any repeat revascularization	83 (3.5)	148 (4.5)	0.071	1.280 (0.978–1.674)	0.072	1.004 (0.698–1.443)	0.983
Hospitalization for HF	27 (1.0)	62 (1.7)	0.033	1.627 (1.035–2.556)	0.035	1.428 (0.760–2.685)	0.268
	**BB + RASI (−)**						
	**Group A2**	**Group B2**		**Unadjusted**		**Adjusted ^b^**	
	**Normoglycemia (*n* = 243)**	**Prediabetes (*n* = 306)**	**Log-Rank**	**HR (95% CI)**	***p* value**	**HR (95% CI)**	***p* value**
MACEs	38 (16.2)	61 (20.6)	0.228	1.281 (1.854–1.921)	0.231	1.544 (0.942–2.532)	0.085
All-cause death	31 (12.8)	55 (18.5)	0.122	1.411 (0.909–2.192)	0.125	1.518 (0.872–2.642)	0.140
Cardiac death	26 (10.8)	48 (16.1)	0.112	1.467 (0.911–2.365)	0.115	1.527 (0.830–2.810)	0.173
Re-MI	5 (2.2)	7 (2.6)	0.831	1.133 (0.359–3.569)	0.832	2.272 (0.522–9.882)	0.274
Any repeat revascularization	7 (3.7)	17 (7.1)	0.139	1.921 (0.797–4.633)	0.146	1.569 (0.419–3.878)	0.504
Hospitalization for HF	6 (2.5)	12 (3.9)	0.343	1.600 (0.600–4.263)	0.347	2.507 (0.786–7.992)	0.120
	**Group A2**	**Group C2**		**Unadjusted**		**Adjusted ^b^**	
	**Normoglycemia (*n* = 243)**	**Diabetes (*n* = 417)**	**Log-Rank**	**HR (95% CI)**	***p* value**	**HR (95% CI)**	***p* value**
MACEs	38 (16.2)	119 (29.1)	<0.001	1.924 (1.335–2.772)	<0.001	1.708 (1.126–2.590)	0.012
All-cause death	31 (12.8)	102 (24.8)	0.001	2.005 (1.341–2.997)	0.001	1.759 (1.109–2.788)	0.016
Cardiac death	26 (10.8)	87 (21.2)	0.001	2.033 (1.312–3.151)	0.002	1.844 (1.106–3.075)	0.019
Re-MI	5 (2.2)	15 (4.6)	0.212	1.885 (0.685–5.187)	0.220	1.573 (0.503–4.923)	0.436
Any repeat revascularization	7 (3.7)	27 (8.6)	0.035	2.378 (1.036–5.462)	0.041	2.044 (0.828–5.045)	0.121
Hospitalization for HF	6 (2.5)	16 (3.8)	0.347	1.562 (0.611–3.993)	0.351	1.227 (0.427–3.821)	0.662
	**Group B2**	**Group C2**		**Unadjusted**		**Adjusted ^b^**	
	**Prediabetes (*n* = 306)**	**Diabetes (*n* = 417)**	**Log-Rank**	**HR × (95% CI)**	***p* value**	**HR (95% CI)**	***p* value**
MACEs	61 (20.6)	119 (29.1)	0.009	1.501 (1.102–2.044)	0.010	1.301 (0.913–1.854)	0.146
All-cause death	55 (18.5)	102 (24.8)	0.034	1.421 (1.024–1.972)	0.036	1.225 (0.839–1.789)	0.293
Cardiac death	48 (16.1)	87 (21.2)	0.069	1.384 (0.973–1.969)	0.071	1.232 (0.814–1.863)	0.323
Re-MI	7 (2.6)	15 (4.6)	0.256	1.673 (0.682–4.103)	0.261	1.564 (0.601–3.666)	0.325
Any repeat revascularization	17 (7.1)	27 (8.6)	0.475	1.247 (0.680–2.288)	0.476	1.077 (0.540–2.148)	0.833
Hospitalization for HF	12 (3.9)	16 (3.8)	0.951	1.024 (0.484–2.164)	0.951	1.404 (0.592–3.331)	0.441

^a^ Adjusted by male, age, LVEF (≤40%), BMI, DBP, STEMI, hypertension, dyslipidemia, previous MI, previous PCI, previous CVA, current smokers, CK-MB, NT-ProBNP, serum creatinine, eGFR (<60 mL/min/1.73 m^2^), total cholesterol, triglyceride, HDL cholesterol, LDL cholesterol, clopidogrel, ticagrelor, cilostazole, CCBs, lipid-lowering agents, LAD (IRA), RCA (IRA and treated vessel), one-vessel disease, ≥3-vessel disease, stent diameter, and number of stents. ^b^ Adjusted by hypertension, current smokers, NT-ProBNP, eGFR (<60 mL/min/1.73 m^2^), triglyceride, LDL cholesterol, and one-vessel disease. BB, ß-blockers; HF, heart failure; LVEF, left ventricular ejection fraction; BMI, body mass index; DBP, diastolic blood pressure; STEMI, ST-elevation myocardial infarction; NSTEMI, non-STEMI; PCI, percutaneous coronary intervention; CVA, cerebrovascular accident; CK-MB, creatine kinase myocardial band; NT-ProBNP, N-terminal pro-brain natriuretic peptide; eGFR, estimated glomerular filtration rate; HDL, high-density lipoprotein; LDL, low-density lipoprotein; CCBs, calcium channel blockers; IRA, infarct-related artery; LAD, left anterior descending coronary artery; RCA, right coronary artery; HR, hazard ratio; CI, confidence interval.

**Table 3 jcm-09-03447-t003:** Comparison of clinical outcomes between BB + RASI users and nonusers at two-year follow-up.

Outcomes	BB + RASI (+) A1 (*n* = 2217)	BB + RASI (−) A2 (*n* = 243)	Log-Rank	Unadjusted		Adjusted ^a^	
				HR (95% CI)	*p* Value	HR (95% CI)	*p* value
Normoglycemia							
MACEs	109 (5.5)	38 (16.2)	<0.001	3.484 (2.408–5.041)	<0.001	3.033 (1.654–5.562)	<0.001
All-cause death	30 (1.5)	31 (12.8)	<0.001	10.16 (6.146–16.78)	<0.001	6.495 (2.820–14.96)	<0.001
Cardiac death	20 (1.0)	26 (10.8)	<0.001	12.63 (7.051–22.64)	<0.001	8.848 (3.121–19.09)	<0.001
Re-MI	30 (1.5)	5 (2.2)	0.299	1.643 (0.637–4.235)	0.304	1.702 (0.458–6.327)	0.427
Any revascularization	58 (3.0)	7 (3.7)	0.579	1.248 (0.570–2.734)	0.580	1.438 (0.298–6.945)	0.651
Hospitalization for HF	17 (0.8)	6 (2.5)	0.008	3.256 (1.284–8.259)	0.013	3.242 (1.279–7.525)	0.014
**Outcomes**	**BB + RASI × (+) B1 (*n* = 2601)**	**BB + RASI (−) B2 (*n* = 306)**	**Log-Rank**	**Unadjusted**		**Adjusted ^b^**	
				**HR (95% CI)**	***p* value**	**HR (95% CI)**	***p* value**
Prediabetes							
MACEs	166 (6.8)	61 (20.6)	<0.001	3.549 (2.646–4.761)	<0.001	3.450 (2.288–4.202)	<0.001
All-cause death	50 (2.0)	55 (18.5)	<0.001	10.34 (7.048–15.17)	<0.001	9.197 (5.246–14.82)	<0.001
Cardiac death	38 (1.5)	48 (16.1)	<0.001	11.68 (7.627–17.88)	<0.001	10.20 (5.322–16.54)	<0.001
Re-MI	40 (1.6)	7 (2.6)	0.203	1.676 (0.751–3.743)	0.208	1.891 (0.666–4.972)	0.175
Any revascularization	83 (3.5)	17 (7.1)	0.005	2.073 (1.230–3.493)	0.006	2.134 (1.314–4.038)	0.005
Hospitalization for HF	27 (1.0)	12 (3.9)	<0.001	3.855 (1.953–7.610)	<0.001	3.487 (1.556–6.234)	0.011
**Outcomes**	**BB + RASI (+) C1 (*n* = 3862)**	**BB + RASI (−) C2 (*n* = 417)**	**Log-Rank**	**Unadjusted**		**Adjusted ^c^**	
				**HR (95% CI)**	***p* value**	**HR (95% CI)**	***p* value**
Diabetes							
MACEs	305 (9.0)	119 (29.1)	<0.001	4.142 (3.351–5.121)	<0.001	4.265 (2.937–6.193)	<0.001
All-cause death	117 (3.4)	102 (24.8)	<0.001	8.959 (6.868–11.69)	<0.001	7.227 (4.374–10.94)	<0.001
Cardiac death	79 (2.3)	87 (21.2)	<0.001	11.06 (8.150–15.00)	<0.001	9.005 (5.156–13.73)	<0.001
Re-MI	86 (2.6)	15 (4.6)	0.022	1.881 (1.087–3.256)	0.024	2.957 (1.380–6.336)	0.005
Any revascularization	148 (4.5)	27 (8.6)	0.001	2.020 (1.340–3.045)	0.001	4.360 (2.344–8.108)	<0.001
Hospitalization for HF	62 (1.7)	16 (3.8)	0.002	2.323 (1.341–4.024)	0.003	2.015 (1.035–3.156)	0.031

^a^ Adjusted by age, LVEF (≤40%), SBP, DBP, CK-MB, NT-ProBNP, eGFR (<60 mL/min/1.73 m^2^), total cholesterol, triglyceride, HDL cholesterol, aspirin, and lipid-lowering agents. ^b^ Adjusted by age, LVEF (≤40%), BMI, SBP, DBP, previous HF, triglyceride, aspirin, CCBs, and lipid-lowering agents. ^c^ Adjusted by age, LVEF (≤40%), SBP, DBP, peak CK-MB, blood glucose, NT-ProBNP, hs-CRP, serum creatinine, eGFR (<60 mL/min/1.73 m^2^), aspirin, cilostazole, lipid-lowering agents, and ACC/AHA type B2 lesion. BB, ß-blockers; RASI, renin–angiotensin system inhibitors; HR, hazard ratio; CI, confidence interval; MACEs, major adverse cardiac events; Re-MI, recurrent myocardial infarction; HF, heart failure; LVEF, left ventricular ejection fraction; BMI, body mass index; SBP, systolic blood pressure; DBP, diastolic blood pressure; CK-MB, creatine kinase myocardial band; NT-ProBNP, N-terminal probrain natriuretic peptide; hs-CRP, high sensitivity C-reactive protein; eGFR, estimated glomerular filtration rate; HDL, high-density lipoprotein; CCBs, calcium channel blockers; ACC/AHA, American College of Cardiology/American Heart Association.

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
