# Peer review of "Beta-Blocker and Renin–Angiotensin System Inhibitor Combination Therapy in Patients with Acute Myocardial Infarction and Prediabetes or Diabetes Who Underwent Successful Implantation of Newer-Generation Drug-Eluting Stents: A Retrospective Observational Registry Study"

_jcm, 2020, doi:10.3390/jcm9113447_

Round 1

Reviewer 1 Report

Overall, this is a well-designed registry-based retrospective cohort study with appropriate safeguards for ascertainment of exposures and outcomes, including an independent event adjudicating committee. The manuscript is well written, and it is easy for practicing physicians with minimum epidemiology skills to understand. Addressing the following items will further strengthen this manuscript.

MAJOR

  1. Almost 40% of the >21,000 patients with AMI were excluded due to incomplete laboratory results including unidentified results of blood hemoglobin A1c and blood glucose, in addition to those excluded due to loss follow up and receiving BB or RASI alone. This appears to be rather large proportion, and can introduce selection bias if those who did not have HbA1c and blood glucose during hospitalization had a different clinical profile than those who have these tests. The authors have already noted incomplete follow-up data in the Discussion section (line 279-281), it is advisable to specifically discuss the potential implications of a large number of exclusions from the cohort on internal and external validity of study results in the Discussion section.

MINOR

  1. The statement in lines 113-114 about leaving the use of triple antiplatelet therapy to the discretion of the individual operators can be confusing to the readers. Were all choices of therapy left to the individual treating physicians since this is an observation study using the registry data and not an experimental study? Was the registry specifically designed for this study?
  2. Were there any differential losses to follow up among different groups (which can potentially introduce bias)?
  3. Is there a biological plausibility of BB+RASI therapy associating with different frequency of Re-MI between prediabetes and diabetes in patients with AMI during a 2-year follow-up period despite similarity in occurrence of other hard outcomes?
  4. It will be easier for the readers to follow if a graphic display is included for BB+RASI users vs non-users’ comparison in addition to the table.

Author Response

Response to Reviewer 1 Comments

First of all, we sincerely thank reviewer for his/her efforts in evaluating our original submission. We also thank the reviewer for the helpful comments and recommendations, which we believe have helped us to improve our manuscript.

Comments and Suggestions for Authors

Overall, this is a well-designed registry-based retrospective cohort study with appropriate safeguards for ascertainment of exposures and outcomes, including an independent event adjudicating committee. The manuscript is well written, and it is easy for practicing physicians with minimum epidemiology skills to understand. Addressing the following items will further strengthen this manuscript.

MAJOR

Point 1: Almost 40% of the >21,000 patients with AMI were excluded due to incomplete laboratory results including unidentified results of blood hemoglobin A1c and blood glucose, in addition to those excluded due to loss follow up and receiving BB or RASI alone. This appears to be rather large proportion, and can introduce selection bias if those who did not have HbA1c and blood glucose during hospitalization had a different clinical profile than those who have these tests. The authors have already noted incomplete follow-up data in the Discussion section (line 279-281), it is advisable to specifically discuss the potential implications of a large number of exclusions from the cohort on internal and external validity of study results in the Discussion section.

Response 1: Thank you for reviewer’s reasonable recommendations. According to reviewer’s recommendation we revised discussion section as follows,

Before, (Line No. 267-281)

Even though there are diverse kinds of newly developed antidiabetic drugs and many efforts to prevent and delay the progression of disease in patients with prediabetes, intensive lifestyle changes and metformin therapy are the only universally accepted interventions for diabetes prevention [37]. BB and RASI are well-known drugs which can reduce the short- and long-term morbidity and mortality in patients with AMI regardless of their glycemic status. However, current guidelines [4-7] do not suggest indications or have not shown the effects of these drugs in patients with AMI and prediabetes, and other reports regarding these comparative results between prediabetes and diabetes in such conditions have not been reported. Therefore, we believe that the results of this study can provide important insights for interventional cardiologists concerning the effects of BB + RASI therapy in patients with AMI and prediabetes and the clinical implication of prediabetes in patients with AMI. Additionally, even though the size of the study population may be insufficient for conclusion, in this nationwide, retrospective, observational, multicenter registry analysis, more than 50 community and teaching hospitals in South Korea participated. Furthermore, larger well-designed randomized controlled trials are required to confirm these results focusing on newer-generation DESs.

After,

Even though there are diverse kinds of newly developed antidiabetic drugs and many efforts to prevent and delay the progression of disease in patients with prediabetes, intensive lifestyle changes and metformin therapy are the only universally accepted interventions for diabetes prevention [37]. BB and RASI are well-known drugs which can reduce the short- and long-term morbidity and mortality in patients with AMI regardless of their glycemic status. However, current guidelines [4-7] do not suggest indications or have not shown the effects of these drugs in patients with AMI and prediabetes, and other reports regarding these comparative results between prediabetes and diabetes in such conditions have not been reported. Therefore, we believe that the results of this study can provide important insights for interventional cardiologists concerning the effects of BB + RASI therapy in patients with AMI and prediabetes and the clinical implication of prediabetes in patients with AMI. In this study, 11877 patients (55.6%) were excluded due to incomplete laboratory results including unidentified results of blood HbA1c and blood glucose, in addition to those excluded due to loss follow up and receiving BB or RASI alone. This appears to be rather large proportion, and can introduce selection bias if those who did not have HbA1c and blood glucose during hospitalization had a different clinical profile than those who have these tests. As mentioned, the baseline (e.g., sex difference, proportion of STEMI, degree of LVEF, renal function) and lesion, and procedural characteristic (e.g., extent of coronary artery disease, length of deployed stent) of the excluded patients could affect the main results of our study. Moreover, the ratio of three groups (normoglycemia, prediabetes, and diabetes) of the excluded patients may act as an important determinant of major clinical outcomes. Therefore, regarding these composition of this study population, caution is needed in interpreting these results. Even though the size of the study population may be insufficient for conclusion, in this nationwide, retrospective, observational, multicenter registry analysis, more than 50 community and teaching hospitals in South Korea participated. Furthermore, larger well-designed randomized controlled trials are required to confirm these results focusing on newer-generation DESs.

MINOR

Point 2: The statement in lines 113-114 about leaving the use of triple antiplatelet therapy to the discretion of the individual operators can be confusing to the readers. Were all choices of therapy left to the individual treating physicians since this is an observation study using the registry data and not an experimental study? Was the registry specifically designed for this study?

Response 2: Thank you for reviewer’s question. We are very sorry; this study was not a randomized controlled study but a retrospective cohort study. Therefore, above our description is incorrect. According to reviewer’s comments, we revised above sentence as follows,

Before, (Line No. 109-116)

2.2. Percutaneous coronary intervention (PCI) procedure and medical treatment

Diagnostic coronary angiography and PCI were done through either the femoral or radial artery using standard techniques [10]. Aspirin (200–300mg) and clopidogrel (300–600mg) were administered before PCI, Alternatively, ticagrelor (180mg) or prasugrel (60mg) was administered. After PCI, the total duration of dual antiplatelet therapy (DAPT; a combination of aspirin [100mg/day] with clopidogrel [75mg/day], ticagrelor [90mg twice daily], or prasugrel [5–10mg/day]) was recommended for > 12 months. Additionally, the use of triple antiplatelet therapy (cilostazol [100 mg twice daily] added to DAPT) was left to the discretion of the individual operators.

After,

2.2. Percutaneous coronary intervention (PCI) procedure and medical treatment

Diagnostic coronary angiography and PCI were done through either the femoral or radial artery using standard techniques [10]. Aspirin (200–300mg) and clopidogrel (300–600mg) were administered before PCI, Alternatively, ticagrelor (180mg) or prasugrel (60mg) was administered. After PCI, the total duration of dual antiplatelet therapy (DAPT; a combination of aspirin [100mg/day] with clopidogrel [75mg/day], ticagrelor [90mg twice daily], or prasugrel [5–10mg/day]) was recommended for > 12 months. Additionally, the use of triple antiplatelet therapy (cilostazol [100 mg twice daily] added to DAPT) was left to the discretion of the individual operators. Moreover, in this registry, there were no operator restrictions for performing PCI.

Point 3: Were there any differential losses to follow up among different groups (which can potentially introduce bias)?

Response 3: Thank you for reviewer’s question. The number of patients who were lost to follow-up was as follows and which may not be act as a bias in this study.

(Figure please see attachment.)

Point 4:  Is there a biological plausibility of BB+RASI therapy associating with different frequency of Re-MI between prediabetes and diabetes in patients with AMI during a 2-year follow-up period despite similarity in occurrence of other hard outcomes?

Response 4: Thank you for reviewer’s question. We think that this point is very important in our study. However, because lack of prior published data regarding this biological plausibility of BB+RASI therapy associating with different frequency of Re-MI between prediabetes and diabetes in patients with AMI, there is a number of limitations in providing very satisfactory information. In the Hoorn Study (Diabetes Care 2013, 36, 3498), the authors showed that the risk of a recurrent cardiovascular event was similar between the normoglycemia and prediabetes groups. However, individuals with diabetes had an increased risk of recurrent cardiovascular events compared to individuals with normoglycemia during a median 4.1 years of follow-up after the first event. According to the recent report (reference No. 30, J. Diabetes Complications 2020, 34, 107574), after adjustment, in RASI users, the cumulative incidence of Re-MI of the diabetes group was significantly higher than that of the prediabetes group (aHR: 1.999; 95% CI: 1.153–3.467; p = 0.014). Hence, as mentioned in discussion section (“Deedwania et al. [31] suggested that the association between diabetes and Re-MI may be a direct effect of diabetes.”), this higher cumulative incidence of Re-MI may be related with the fundamental characteristics of diabetes. Therefore, we revised discussion section including this content as follows,

Before, (Line No. 247-253)

Deedwania et al. [31] suggested that the association between diabetes and Re-MI may be a direct effect of diabetes. Hyperglycemia has been linked to an increase in plasma renin activity with RAS activation, which is known to impair insulin signaling [32]. Hyperglycemia, insulin resistance, and advanced glycation end-products can contribute to vascular inflammation and endothelial dysfunction in patients with diabetes [33]. Increased platelet activation, the presence of a chronic hypercoagulable state, and impaired fibrinolysis are also considerable factors in this unfavorable clinical outcome [34].

After,

In the Hoorn Study [31], the risk of a recurrent cardiovascular event was similar between the normoglycemia and prediabetes groups. However, individuals with diabetes had an increased risk of recurrent cardiovascular events compared to individuals with normoglycemia during a median 4.1 years of follow-up after the first event. According to the recent report [30], after adjustment, in RASI users, the cumulative incidence of Re-MI of the diabetes group was significantly higher than that of the prediabetes group (aHR: 1.999; 95% CI: 1.153–3.467; p = 0.014). Deedwania et al. [32] suggested that the association between diabetes and Re-MI may be a direct effect of diabetes. Hyperglycemia has been linked to an increase in plasma renin activity with RAS activation, which is known to impair insulin signaling [33]. Hyperglycemia, insulin resistance, and advanced glycation end-products can contribute to vascular inflammation and endothelial dysfunction in patients with diabetes [34]. Increased platelet activation, the presence of a chronic hypercoagulable state, and impaired fibrinolysis are also considerable factors in this unfavorable clinical outcome [35].

Because we newly added reference number 31 in above sentence, the number of the references was re-numbered as follows.

31 → 32, 32 → 33, 33 → 34, 34 → 35, 35 → 36, 36 → 37, 37 → 38, 39 → 40, 40 → 41, 41 → 42, 42 → 43.

Point 5:  It will be easier for the readers to follow if a graphic display is included for BB+RASI users vs non-users’ comparison in addition to the table.

Response 5: (Line No. 205) Thank you for reviewer’s recommendation. We totally agree with reviewer’s recommendation and we revised Figure 2 as follows. In this revised Figure 2, all patients who received BB+RASI or not are included. Therefore, Supplementary file (Figure 3) is deleted. Moreover, Supplementary file (Figure 4) is renamed as Supplementary File (Figure 3).

(Figures please see attachment.)

Before,

3.2. Clinical outcomes (Line No. 179-203)

Table 2 and 3and Fig. 2 show the cumulative incidences of major clinical outcomes during the 2-year follow-up period. In Table 2, the cumulative incidence of MACEs (adjusted hazard ratio [aHR]: 1.127; 95% CI: 0.860-1.477; p = 0.387), all-cause death, CD, any repeat revascularization, and HHF (aHR: 1.428; 95% CI: 0.760-2.685; p = 0.268) were similar between group B1 (prediabetes and BB + RASI users) and C1 (diabetes and BB+RASI users). However, the cumulative incidence of Re-MI was higher in group C1 than group B1 (aHR: 1.660; 95% CI: 1.000-2.755; p = 0.020). The cumulative incidences of MACEs (aHR: 1.464; 95% CI: 1.022–2.096; p = 0.038) was higher in group B1 than in group A1 (normoglycemia and BB+RASI users). The cumulative incidence of MACEs (aHR: 1.587; 95% CI: 1.139–2.012; p = 0.006) and Re-MI (aHR: 2.275; 95% CI: 1.218–4.247; p = 0.010) were significantly higher in group C1 than in group A1. In BB + RASI non-users, the cumulative incidences of MACEs (aHR: 1.708; 95% CI: 1.126-2.590; p = 0.012), all-cause death (aHR: 1.759; 95% CI: 1.109-2.788; p = 0.016), and CD (aHR: 1.844; 95% CI: 1.106-3.075; p = 0.019) were higher in group C2 (diabetes) than in group A2 (normoglycemia). In Table 3, in all three groups (normoglycemia, prediabetes, and diabetes), BB+RASI therapy reduced the cumulative incidences of MACEs, all-cause death, CD, and HHF. Additionally, in the prediabetes group, BB+RASI users showed lower cumulative incidences of any repeat revascularization than that of BB+RASI non-users. In the diabetes group, BB+RASI users showed lower cumulative incidences of Re-MI and any repeat revascularization than those of BB+RASI non-users. Kaplan-Meier analyses of major clinical outcomes of BB+RASI non-users are shown in Supplementary file 3. Independent predictors for MACEs and HHF in BB+RASI users at 2 years are shown in Supplementary file 4. Male sex, STEMI, decreased LVEF (< 40%), lipid-lowering agents, decreased eGFR (< 60 mL/min/1.73m2), ≥ 3-vessel disease, the use of IVUS, and ≥ 30 mm length of the deployed stent were meaningful independent predictors for MACEs. Moreover, old age (≥ 65 years), STEMI, decreased LVEF, decreased eGFR, and ACC/AHA type B2/C lesions were independent predictors for HHF in this study.

After,

3.2. Clinical outcomes

Table 2 and 3and Fig. 2 show the cumulative incidences of major clinical outcomes during the 2-year follow-up period. In Table 2, the cumulative incidence of MACEs (adjusted hazard ratio [aHR]: 1.127; 95% CI: 0.860-1.477; p = 0.387), all-cause death, CD, any repeat revascularization, and HHF (aHR: 1.428; 95% CI: 0.760-2.685; p = 0.268) were similar between group B1 (prediabetes and BB + RASI users) and C1 (diabetes and BB+RASI users). However, the cumulative incidence of Re-MI was higher in group C1 than group B1 (aHR: 1.660; 95% CI: 1.000-2.755; p = 0.020). The cumulative incidences of MACEs (aHR: 1.464; 95% CI: 1.022–2.096; p = 0.038) was higher in group B1 than in group A1 (normoglycemia and BB+RASI users). The cumulative incidence of MACEs (aHR: 1.587; 95% CI: 1.139–2.012; p = 0.006) and Re-MI (aHR: 2.275; 95% CI: 1.218–4.247; p = 0.010) were significantly higher in group C1 than in group A1. In BB + RASI non-users, the cumulative incidences of MACEs (aHR: 1.708; 95% CI: 1.126-2.590; p = 0.012), all-cause death (aHR: 1.759; 95% CI: 1.109-2.788; p = 0.016), and CD (aHR: 1.844; 95% CI: 1.106-3.075; p = 0.019) were higher in group C2 (diabetes) than in group A2 (normoglycemia). In Table 3, in all three groups (normoglycemia, prediabetes, and diabetes), BB+RASI therapy reduced the cumulative incidences of MACEs, all-cause death, CD, and HHF. Additionally, in the prediabetes group, BB+RASI users showed lower cumulative incidences of any repeat revascularization than that of BB+RASI non-users. In the diabetes group, BB+RASI users showed lower cumulative incidences of Re-MI and any repeat revascularization than those of BB+RASI non-users. Kaplan-Meier analyses of major clinical outcomes of BB+RASI non-users are shown in Supplementary file 3. Independent predictors for MACEs and HHF in BB+RASI users at 2 years are shown in Supplementary file 3. Male sex, STEMI, decreased LVEF (< 40%), lipid-lowering agents, decreased eGFR (< 60 mL/min/1.73m2), ≥ 3-vessel disease, the use of IVUS, and ≥ 30 mm length of the deployed stent were meaningful independent predictors for MACEs. Moreover, old age (≥ 65 years), STEMI, decreased LVEF, decreased eGFR, and ACC/AHA type B2/C lesions were independent predictors for HHF in this study.

Reviewer 2 Report

Congratulations for the excellent work. Although it was hard to go through the enormous amount of data, they support the conclusions. The major limitation, however clearly stated by the authors, is the lack of details on the adherence to the therapy prescribed. It should be emphasized in the conlusion that the results should be interpreted with caution due to the lack of feeback on the compliance.

I have to minor remarks:

  • please comment on the multiple comparisons performed and the statistical approach to avoid the look-elsewhere effect,
  • what is the possible explanations fo the IVUS contribution to the increased MACE incidence (as presented in the suppl. table)?

Author Response

Response to Reviewer 2 Comments

First of all, we sincerely thank reviewer for his/her efforts in evaluating our original submission. We also thank the reviewer for the helpful comments and recommendations, which we believe have helped us to improve our manuscript.

Comments and Suggestions for Authors

Point 1: Congratulations for the excellent work. Although it was hard to go through the enormous amount of data, they support the conclusions. The major limitation, however clearly stated by the authors, is the lack of details on the adherence to the therapy prescribed. It should be emphasized in the conlusion that the results should be interpreted with caution due to the lack of feeback on the compliance.

Response 1: Thank you for reviewer’s comments and recommendations. According to reviewer’s recommendations, we revised the “Conclusion” section as follows,

Before, (Line No. 313-317)

  1. Conclusions

In conclusion, in this study, prediabetic and diabetic patients are not different in their outcome when treated with BB and RASI over 2 years but they certainly both are better off than the corresponding groups without treatment with these established drugs post-AMI. However, further well-designed studies are warranted to confirm these results.

After,

  1. Conclusions

In conclusion, in this study, prediabetic and diabetic patients are not different in their outcome when treated with BB and RASI over 2 years but they certainly both are better off than the corresponding groups without treatment with these established drugs post-AMI. However, the results should be interpreted with caution due to the lack of details on the adherence and feedback to the therapy prescribed. Moreover, further well-designed studies are warranted to confirm these results.

I have to minor remarks:

Point 2: please comment on the multiple comparisons performed and the statistical approach to avoid the look-elsewhere effect

Response 1: Thank you for reviewer’s comments and recommendations. We revised “2.4. Statistical analysis” section as follows,

Before, (Line No. 134-145)

2.4. Statistical analysis

For continuous variables, differences among the three groups were evaluated using an analysis of variance or the Jonckheere-Terpstra test, while a post-hoc analysis was performed using the Hochberg test or Dunnett T3 test; the data are expressed as the mean±standard deviation. For categorical variables, intergroup differences were analyzed using a χ2 test or Fisher’s exact test, as appropriate. Data are expressed as counts and percentages. Various clinical outcomes were estimated using the Kaplan-Meier method, and intergroup differences were compared using a log-rank test. Any variables with a p value of <0.001 in the univariate analysis and conventional risk factors of poor outcomes in the AMI population were considered potential confounding factors, and were entered into the multivariate analysis. (Table 2, Table 3). For all analyses, two-sided p-values <0.05 were considered statistically significant. All statistical analyses were performed using SPSS version 20 (IBM; Armonk, NY, USA).

After,

2.4. Statistical analysis

For continuous variables, differences among the three groups were evaluated using an analysis of variance or the Jonckheere-Terpstra test, while a post-hoc analysis was performed using the Hochberg test or Dunnett T3 test; the data are expressed as the mean±standard deviation. For categorical variables, intergroup differences were analyzed using a χ2 test or Fisher’s exact test, as appropriate. Data are expressed as counts and percentages. Various clinical outcomes were estimated using the Kaplan-Meier method, and intergroup differences were compared using a log-rank test. To determine meaningful variables, all variables with p <0.1 were included in the univariate analysis. After univariate analysis, variables with p <0.001 and known conventional risk factors of poor outcomes in the AMI population were considered potential confounding factors, and were entered into the multivariate analysis. These variables included the following: male, age, LVEF (≤40%), body mass index, diastolic blood pressure, ST-segment-elevation myocardial infarction (STEMI), hypertension, dyslipidemia, previous MI, PCI, and cerebrovascular accidents, current smokers, creatine kinase myocardial band (CK-MB), N-terminal pro-brain natriuretic peptide (NT-ProBNP), serum creatinine, eGFR (<60mL/min/1.73m2), total cholesterol, triglyceride, high-density lipoprotein-cholesterol, low-density lipoprotein-cholesterol, clopidogrel, ticagrelor, cilostazole, calcium channel blockers, lipid lowering agents, left anterior descending coronary artery (LAD, IRA), right coronary artery (RCA, IRA and treated vessel), 1-vessel disease, ≥ 3-vessel disease, stent diameter, and number of stent (Table 2). In the same way, the comparisons of major clinical outcomes between BB + RASI users and non-users in all three groups (normoglycemia, prediabetes, and diabetes) were performed as shown in Table 3. For all analyses, two-sided p-values <0.05 were considered statistically significant. All statistical analyses were performed using SPSS version 20 (IBM; Armonk, NY, USA).

Point 3: what is the possible explanations fo the IVUS contribution to the increased MACE incidence (as presented in the suppl. table)?

Response 1: Thank you for reviewer’s comments and recommendations. As reviewer said, we could not find any evidence that IVUS contributed to the increased MACE incidence through this paper. Moreover, unadjusted HR of IVUS for MACE was incorrect. The exact unadjusted HR of IVUS for MACE is 1.120 (0.951-1.320, p = 0.176). Maybe there were some mistakes during the calculation at that time. This is entirely our fault. We sincerely apologize for our mistakes. Please count it generously. After a review, we find that other values are correct, except for the value of  IVUS. Hence, we revised Supplement Table S3 (renamed Supplement Table S4) as follows,

Before,

 (Supplementary table please see attachment.)

After,

(Supplementary table please see attachment.)
